# Statistical Analysis and Forecasts of Performance Indicators in the Romanian Healthcare System

**DOI:** 10.3390/healthcare13020102

**Published:** 2025-01-07

**Authors:** Cristian Ovidiu Drăgan, Laurențiu Stelian Mihai, Ana-Maria Camelia Popescu, Ion Buligiu, Lucian Mirescu, Daniel Militaru

**Affiliations:** Faculty of Economics and Business Administration, University of Craiova, 200585 Craiova, Dolj, Romania; cdragano11@gmail.com (C.O.D.); popescu.ana.g6w@student.ucv.ro (A.-M.C.P.); ion.buligiu@edu.ucv.ro (I.B.); mirescu.lucian.m5t@student.ucv.ro (L.M.); daniel.militaru@edu.ucv.ro (D.M.)

**Keywords:** healthcare, performance, forecast, KPI, ARIMA, COVID-19, statistical analysis, hospital performance, healthcare system

## Abstract

Background/Objectives: Globally, healthcare systems face challenges in optimizing performance, particularly in the wake of the COVID-19 pandemic. This study focuses on the analysis and forecasting of key performance indicators (KPIs) for the County Emergency Clinical Hospital in Craiova, Romania. The study evaluates indicators such as average length of stay (ALoS), bed occupancy rate (BOR), number of cases (NC), case mix index (CMI), and average cost per hospitalization (ACH), providing insight into their dynamics and future trends. Methods: We performed statistical analyses on quarterly data from 2010 to 2023, employing descriptive statistics and stationarity tests (e.g., Dickey–Fuller), using ARIMA models to forecast each KPI, ensuring model validation through tests for autocorrelation, heteroscedasticity, and stationarity. The model selection prioritized Akaike and Schwarz criteria for robustness. Results: The findings reveal that ALoS and BOR demonstrate seasonality and are influenced by colder months, and it is expected that the ALoS will stabilize to around five days by 2025. Moreover, we predict that the BOR will range between 46 and 52%, reflecting these seasonal variations. The NC forecasts indicate a post-pandemic recovery but to below pre-pandemic levels, and we project the CMI to stabilize at around 1.54, suggesting a return to consistent case complexity. The ACH showed significant growth, particularly in the fourth quarter, driven by inflation and seasonal costs, and it is projected to reach more than RON 3000 by 2025. Conclusions: This study highlights the utility of ARIMA models in forecasting healthcare KPIs, enabling proactive resource planning and decision-making. The findings underscore the impact of seasonality and economic factors on hospital operations, offering valuable insights for improving efficiency and adapting to post-pandemic challenges.

## 1. Introduction

Health is an important component of socio-economic progress since healthy individuals contribute to economic development. In this context, healthcare systems enhance overall public health and address populations’ medical needs [1,2]. Over recent decades, advances in medical knowledge and technology, as well as increased public awareness, have contributed to an image of healthcare systems as a complex concept, which includes several physical, social, and economic dimensions [3,4]. Thus, healthcare systems are experiencing significant transformations in both clinical practices and regulations aiming to improve efficiency and accountability. These changes have led to the development of performance indicators that measure a system’s success, quality, and efficacy [5,6]. In this context, several national and international initiatives have been implemented in order to promote quality and operational effectiveness, as well as to standardize assessments of healthcare services [7,8,9,10].

Current performance measurement systems rely on multiple variables, and challenges exist in optimizing these variables because of administrative issues [11]. While many frameworks focus on improving specific healthcare processes, they often overlook the needs of the larger system. Therefore, it is essential to identify the most impactful and effective key performance indicators (KPIs) that can help in overcoming the challenges facing healthcare systems [5]. These KPIs, often presented as time-series data, provide important information for managing healthcare stress, especially in difficult situations, such as the COVID-19 pandemic.

For an emergency county hospital, it is very important to be able to analyze and accurately forecast KPIs, such as average length of stay (ALoS), bed occupancy rate (BOR), number of cases (NC), case mix index (CMI), and average cost per hospitalization (ACH), especially for the early identification of resource constraints, shift optimization, and preparation for potential increases demand [12]. Moreover, healthcare KPIs are useful for defining and measuring performance, as well as for setting organizational objectives [13,14,15,16].

These indicators are used and reported by all hospitals in Romania, giving them a national character. They are also used, alongside other indicators, by other countries in national and international assessments, as they are included in databases such as EUROSTAT.

Our first KPI, the average length of stay (ALoS), indicates the efficiency of medical case management, which impacts both the cost and availability of beds. Shorter stays while maintaining the integrity of medical records may indicate the optimization of medical procedures and clinical treatments. Analyzing the dynamics of this indicator can provide insight into improvements in medical protocols and resource allocation while also identifying issues that need to be solved. Moreover, medium-term forecasts can contribute to improving strategies and protocols and reducing costs without compromising the quality of care.

Moving on to our second KPI, the bed occupancy rate (BOR) refers to the proportion of occupied beds out of the total available, thus highlighting the hospital’s capacity to meet inpatient demand [17]. This indicator may be influenced by seasonality, fluctuations in case numbers, and the case mix index, and its optimization is essential for preventing overload and ensuring efficient resource management [10]. Understanding annual and seasonal trends in BOR enables a hospital to adjust capacity and manage resources in order to be able to meet demand, while forecasting helps in preparing for peak periods, such as winter months, to prevent overcrowding [12].

Moreover, the evolution in the number of cases (NC) reflects both the demand for healthcare services and their accessibility, which is influenced by external factors such as pandemics or safety measures [18]. Studying this indicator helps in predicting demand, and accurately forecasting the number of cases can assist with capacity planning, shift optimization, and management of patient flow based on anticipating periods of increased inpatient admissions [12]. Moreover, even though exceptional events, such as the pandemic, have had substantial impacts on this indicator, its long-term analysis still provides valuable insights [4].

The fourth indicator, the case mix index (CMI), highlights the complexity and severity of treated cases, reflects the demand for medical expertise and specialized equipment [10], and facilitates an assessment of the quality of care and the hospital’s level of specialization [19]. Analyzing and forecasting this indicator helps the hospital allocate appropriate resources for complex cases, prepares staff to care for patients with serious conditions [20], and, at the same time, supports strategic decisions related to specialized equipment and facilities [12].

Last, our next KPI, the average cost per hospitalization (ACH), is influenced by the case mix index and external economic factors, such as inflation and seasonal expenses [21,22], its management and monitoring being essential to maintaining the hospital’s financial sustainability, especially in the current context, marked by budgetary constraints and inflation. Statistical analysis of its past evolution may provide a solid base for budget management and cost strategy adjustments [23], while forecasting can highlight periods of rising costs, supporting more efficient budget planning [21].

The retrospective statistical analysis (2010–2023), as well as forecasting the upcoming (2024–2025) evolution of these KPIs, are crucial for enhancing the performance and service quality of a county emergency hospital. At the same time, observing the dynamics and the trends of these indicators enables a proactive approach to optimizing medical services, resource planning, and cost management [12], and thus, the hospital can significantly improve its capacity to meet demand while ensuring sustainability and operational efficiency.

The forecasting model used in this paper, ARIMA, is based on well-founded statistical models that enable effective data interpretation. Its parameters—autoregression, integration, moving average—have clear meanings, thus allowing for a better understanding and interpretation of the results. Because of its relatively easy-to-understand concepts, ARIMA can be implemented quickly and efficiently and is useful in interdisciplinary research, being more accessible to readers. ARIMA is very effective for univariate time series, with models based on autocorrelation, and requires a relatively small dataset for calibration, unlike machine-learning models, which often require large volumes of data for robust results. The ARIMA model identifies moderate seasonality in the analyzed data, while the extended SARIMA model can be used with high seasonal data. Machine-learning models can overcome the limitations of ARIMA in capturing complicated relationships or working with multivariate data; however, this does not apply to the data examined in our study. Thus, the choice of ARIMA in this work is scientifically justified, offering high interpretability and efficiency.

In this context, our study aims to explore the evolution and trends of these KPIs, based on data from the County Emergency Clinical Hospital (SCJU Craiova), Craiova, Romania, spanning 15 years, offering a thorough image of their dynamics, through statistical modeling techniques. The paper is structured as follows: the second section presents a review of similar studies, focusing on their methodology, results, and conclusions, while the third section details the research methodology, discussing the statistical methods used to validate the ARIMA forecasting methods. Moving on, the fourth section presents and discusses the study’s results, highlighting the novelty and the original contributions of our paper and concluding with final remarks, limitations, and potential directions for further development.

The novelty of this research lies in the use of ARIMA models to forecast healthcare KPIs for a county emergency hospital in Romania. In our research, we used advanced tests (Dickey–Fuller, Shapiro–Wilk, Durbin–Watson, Breusch–Godfrey, ARCH) to assess the stationarity, normality, and validity of the forecast models, thus ensuring the robustness of the forecasts. Furthermore, multiple ARIMA models are applied to each KPI to provide accurate forecasts and statistics. This allows for the anticipation of the ALoS decreases, BOR stabilization, and a slight increase in the CMI, an approach that allows for predictive management of healthcare demand and resource pressures.

Based on these elements, we consider that the study makes a significant contribution to the field of predictive analysis and healthcare management by integrating robust forecasting models and adapting to the new post-pandemic context, both of these being relevant to enhancing the performance of the healthcare system.

## 2. Literature Review

Healthcare performance assessment is a field that is currently expanding as a response to global changes and new healthcare perspectives. Entities such as the OECD, WHO, and European Union have developed several performance measurement frameworks, integrating both financial and non-financial metrics [24]. These frameworks aim to assess the healthcare system holistically, understanding that organizational success is multi-dimensional and often influenced by broader socio-economic factors [25,26].

Some scholars argue that an efficient healthcare system needs to meet several conditions in terms of availability, quality, fair financial burden, equal access, and social acceptability [20,27]. Performance assessment aligns the system with these conditions, using indicators such as mortality rates, surgical infection rates, and length of stay, which act as procedural efficiency metrics [28].

Moreover, Soyiri [29] argues that health performance forecasting has diverse applications, from resource allocation in emergency departments to chronic disease management linked to environmental factors. However, several issues arise, particularly regarding data limitations and low model accuracy during extreme events, such as sudden spikes in cases. To address these issues, quantile regression, fractional polynomial models, and other probabilistic approaches can be used to improve forecasts in uncertain situations.

There are several scholars who study healthcare system performance at global, regional, and national levels, such as Sun et al. [30], who, based on data from 173 countries, found an average efficiency of national healthcare systems of 78.9%, ranging from 67% in African countries to 86% in West Pacific countries. Moreover, their study also found a significant correlation between national development, HIV/AIDS incidence, governance, and health insurance systems, on the one hand, and the healthcare system’s performance. On the other hand, a 1% increase in social security spending led to a 1.9% rise in system efficiency. In another study, Bitton et al. [31] looked at 14 high-performing low- and middle-income countries, concluding that most of them improved or at least maintained their performance regarding healthcare service quality and health insurance accessibility.

In another study, Saleeshya and Harikumar [32] tested and validated a lean framework for hospital performance in India, using indicators related to areas such as service quality and delivery, medical efficiency and accessibility, and patient-focused practices. In their paper, the scholars studied financial performance metrics such as average revenue per occupied bed, earnings before interest, tax, depreciation, and amortization, and operational revenue using Confirmatory Factor Analysis and Structural Equation Modeling. Their findings suggest that while there is a significant positive correlation between the management’s commitment to lean practices and operational and financial outcomes, improvements in technology and business processes, as well as the stakeholders involved, did not directly or significantly influence the hospital’s performance, measured through the above-mentioned metrics. Thus, they concluded that while practices led to better service delivery and quality, there was no significant correlation between service efficiency, accessibility, or patient-centered practices, on the one hand, and financial success, on the other hand.

Along the same lines, Ghasemzadeh et al. [4] studied the impact that the Health Transformation Plan had on hospitals affiliated with the Qazvin University of Medical Sciences, Iran. They used data ranging from 2012 to 2019, covering both general hospital metrics as well as specific KPIs, such as monthly outpatient visits, inpatient admissions, paraclinical patients, or the number of surgeries. Their findings showed that after the Health Transformation Plan was implemented, the hospitals registered significant increases in outpatient visits, paraclinical cases, and inpatient admissions, concluding that well-designed policies can improve key performance metrics.

Moving on, Stepovic [33] conducted a comparative analysis and forecast of several KPIs related to healthcare staff and medical technology across several Eastern European countries. They studied the trends of these indicators and projected the future demand for staff and equipment throughout 2025, using linear regression and other descriptive and predictive statistical methods. Their findings indicate a steady increase in the demand for healthcare professionals and technology across most countries. Specifically, Latvia is expected to see a significant increase in general practitioners, while Romania and Bulgaria might experience slight declines. Moreover, the number of pharmacists and CT and MRI units is projected to grow in most countries, except for Albania and Bulgaria, which showed potential decreases.

Other scholars, such as Si et al. [5], proposed a hybrid multiple-criteria decision-making model in order to identify relevant KPIs for hospital performance assessment. Thus, through a combination of evidential reasoning, linguistic variables, and the decision-making trial and evaluation laboratory (DEMATEL) approach, their model provided a networked view of performance metrics. According to this model, “nosocomial infections”, “incidents/errors”, “number of operations/procedures”, “length of stay”, “bed occupancy”, and “financial measures” were found to be significant metrics for measuring hospital performance.

Along the same lines, Yadav [34] examines the potential of machine-learning algorithms to forecast the effects of healthcare policies on several social or economic indicators. In this regard, the study demonstrates how big data analytics and machine-learning algorithms can help identify patterns within large sets of data, providing useful insights for policymakers and healthcare professionals. Using historical data, the model trains algorithms to predict the impact of new policies on healthcare performance indicators, such as mortality rates and healthcare access. Findings reveal a positive correlation between healthcare investment and economic growth, with healthcare access linked to GDP per capita, indicating income-based disparities. Moreover, the study also addresses challenges in big data analytics, such as quality and privacy concerns, while highlighting machine learning’s role in healthcare policy reform, hospital performance development, and sustainable economic growth.

Several authors [12,22,23,35,36,37,38,39] agree that there is currently an increased interest in using time-series models to forecast demand for medical services, including patient arrivals, hospital discharges, and length of stay. They show that time-series models such as the Autoregressive Integrated Moving Average (ARIMA) are more efficient than traditional regression since they are able to capture short-term variations, making them ideal for forecasting demand metrics such as patient arrivals, hospital discharges, and length of stay. Studies demonstrate the model’s effectiveness in estimating workloads for emergency departments and other hospital settings, highlighting its value in operational planning [17,40]. At the same time, Zhang [41] found that this model has shown remarkable accuracy in forecasting the spread of infectious illnesses. Along the same lines, a literature review by Grøntved et al. [42] examined several statistical models used to forecast healthcare capacity requirements, finding that 32% of the studies included in their sample were using ARIMA. However, the researcher also noted that not all models were successfully validated or monitored for post-implementation, pointing to the need for ongoing evaluation to ensure the model’s relevance.

The onset of the COVID-19 pandemic prompted extensive application of ARIMA and hybrid models for forecasting case counts and hospital resource needs [12]. Authors such as Darapaneni et al. [43], Hernandez-Matamoros et al. [44], Somyanonthanakul et al. [45], and Wang et al. [46] have demonstrated ARIMA’s accuracy in modeling short-term epidemic outbursts, with variants like the ARIMAX model being able to integrate external factors for more accurate forecasting during epidemics or pandemics. Moreover, other authors such as Abonazel and Darwish [47], Aditya Satrio et al. [48], Gecili et al. [49], Singh et al. [50], and Zrieq et al. [51] have shown that ARIMA models are able to accurately forecast COVID-19 cases, fatalities, and recoveries based on daily reported data. Along the same lines, Kulshreshtha and Garg [18] have shown that ARIMA models outperformed other auto regression models in forecasting new COVID-19 cases in India.

In another study, Duarte [12] studied forecasting models for emergency department indicators in order to assess their capacity for predicting resource demands and mitigating the pandemic’s impact on the department’s KPIs. The study evaluates three methods (ARIMA, Prophet, and General Regression Neural Network) used for projecting patient numbers, visit frequency, admission waitlists, and discharges. Their findings showed that General Regression Neural Networks were better suited to deal with unpredictable fluctuations, outperforming ARIMA and Prophet, yielding lower error rates across all indicators, and being more reliable for use during the pandemic.

Moreover, Soyiri [29] argues that forecasting plays an essential role in anticipating healthcare service demand and supports preventive care and intervention strategies while helping healthcare providers manage risks and prepare for increased demand. The study provides a structured framework that identifies healthcare needs, defines indicators, analyzes historical data, and builds predictive models based on time-series models, such as ARIMA, which are shown to be more efficient in mitigating seasonal fluctuations and long-term trends in the health sector.

Similar to our research, a study by Yardımcı and Boğar [21] uses ARIMA for healthcare forecasting and develops a trend-residual decomposition model in order to project Türkiye’s total healthcare spending based on historical data from 1999 to 2021. The findings proved that ARIMA has superior modeling and forecasting capabilities compared to other models, being able to effectively anticipate Türkiye’s total healthcare spending.

Even though many studies use statistical analysis models and KPI forecasting in the healthcare field, not many of them address this topic on a single hospital, rather the entire national system, and even fewer conduct this type of research in Romania. Thus, in order to bridge this gap, our paper aims to analyze the KPIs of a County Emergency Clinical Hospital in Romania (SCJU Craiova) based on historical quarterly data from 2010 to 2023 and to forecast their evolution for 2024 and 2025 using the ARIMA time-series model.

## 3. Materials and Methods

Our paper aims to analyze and forecast the evolution of the healthcare system with statistical models based on KPIs such as average length of stays (ALoS), bed occupancy rate (BOR), number of cases (NC), case mix index (CMI) and average cost per hospitalization (ACH). Thus, we will use statistical analysis to study data from the County Emergency Clinical Hospital (SCJU Craiova) in Craiova, Romania, for a period of 14 years (2010–2023) and forecast its evolution for the next two years (2024–2025). These variables are presented in Table 1.

Our statistical analysis was based on calculating and interpreting indicators such as mean, variance, standard deviation, and coefficient of variation while using several statistical tests to validate the models. In this regard, the Shapiro–Wilk test [52] is a statistical test used to assess whether a data set follows a normal distribution, as we can see in the following Equation (1):(1)$$W=\frac{(\sum_{i=1}^{n}a_{i}x_{\left(i\right)}{)}^{2}}{\sum_{i=1}^{n}{(x}_{i}-\bar{x}{)}^{2}}$$
where $x_{\left(i\right)}$ represents the ordered data points (from smallest to largest), while $a_{i}$ are specific coefficients of the Shapiro–Wilk test, calculated using the mean and variance of a normal distribution. Moreover, $x_{i}$ represents the individual values of the dataset while $\bar{x}$ is the sample’s mean. The test statistic, *W*, ranges from 0 to 1, indicating the degree of similarity between the observed data distribution and a normal distribution. If *W* is closer to 1, the data are likely to follow a normal distribution, while a significantly lower *W* indicates a possible deviation from normality. The Shapiro–Wilk test is based on the following two hypotheses: the null hypothesis (H_0_), which implies that the data are normally distributed, and the alternative hypothesis (H_1_), which assumes that the data are not normally distributed. If the test’s *p*-value is lower than a certain threshold (usually α = 0.05), the null hypothesis is rejected, suggesting that the data do not follow a normal distribution.

Moreover, the Dickey–Fuller test [53] is used to assess the data’s stationarity, meaning if the statistical properties of the series, such as mean and variance, are constant over time. This characteristic is very important since several models, including ARIMA, assume that data are stationary. If the data are not stationary (a unit root is present in the time series), it can become stationary through differencing. By augmenting the test, additional lags are included to capture complex dependencies within the series. The test is based on the following two hypotheses: the null hypothesis (H_0_), which implies the time series has a unit root, meaning it is non-stationary, and the alternative hypothesis (H_1_), based on the assumption that the time series is stationary (no unit root is present in the data). Similar to the Shapiro–Wilk test, if the *p*-value of the test is below a specified significance level (typically α = 0.05), the null hypothesis is rejected, indicating that the time series is stationary.

The Durbin–Watson test [54] is the third test used for statistical analysis in this paper, and it calculates a variable (*DW*) based on the differences between the residual values of consecutive time points, as seen in the following Equation (2):(2)$$DW=\frac{\sum_{i=2}^{n}{\left({\hat{\varepsilon }}_{i}-{\hat{\varepsilon }}_{i-1}\right)}^{2}}{\sum_{i=1}^{n}{\left({\hat{\varepsilon }}_{i}\right)}^{2}}$$
where *n* represents the number of observations, and *DW* ranges between 0 and 4. A value closer to 0 indicates a higher degree of positive autocorrelation, a value closer to 4 indicates a higher degree of negative autocorrelation, and a value around 2 suggests the absence of first-order autocorrelation. Similar to the previous two tests, the Durbin–Watson test is based on the following two hypotheses: the null hypothesis (H_0_), which implies that there is no first-order autocorrelation (errors are independent), and the alternative hypothesis (H_1_), which assumes that there is a first-order autocorrelation (positive or negative). To assess the result, the *DW* statistic is compared against the lower bound (*d_L_*) and upper bound (*d_U_*) from the Durbin–Watson table for a given significance level, typically 5%. If the *DW* statistic falls within the interval (*d_U_*, 4 − *d_U_*), the null hypothesis (H_0_) is accepted, indicating that the errors are not autocorrelated.

Moving on, the Breusch–Godfrey test [55,56] is a statistical test used to assess whether a regression model’s residuals are higher-order autocorrelated (autocorrelated across multiple lags) in the residuals of a regression model. This test extends the Durbin–Watson test, which is limited to detecting only first-order autocorrelation (i.e., between consecutive residuals). The Breusch–Godfrey test produces an associated *p*-value, which, if greater than 0.05, the null hypothesis (H_0_) is accepted, indicating no evidence of autocorrelation at the chosen significance level of 5%.

The last statistical test used in our study, the ARCH test [57], is based on estimating a regression model and analyzing its residuals to check for dependency in the squared errors, which indicates whether the errors follow an autoregressive pattern. Similarly to the previous tests, ARCH is based on the following two hypotheses: the null hypothesis (H_0_), which implies that there is no conditional heteroscedasticity, meaning the variance of the residuals is constant over time, and the alternative hypothesis (H_1_), which assumes that the time series has a conditional heteroscedasticity, meaning the variance of the residuals depends on their previous values. The ARCH test also calculates a *p*-value, which, if greater than 0.05, implies the validation of the null hypothesis, suggesting that there is no evidence of heteroscedasticity at the 5% significance level.

Regarding the forecast of the performance indicators, our research is using the Autoregressive Integrated Moving-Average (ARIMA) model, which is one of the most widely used techniques for time-series modeling and forecasting [58]. This method has been proven to be efficient at modeling and predicting time-series data that show autocorrelation, meaning there are dependency relationships between past and future values of a variable.

The ARIMA model is characterized by the following three components:-The autoregressive (AR) component assumes that the values are linearly dependent on previous values. In an AR(*p*) model, *p* represents the number of lagged (previous) terms used to predict the current value. The general formula for the AR component is seen in the following Equation (3):(3)$$X_{t}=c+\phi_{1}X_{t-1}+\phi_{1}X_{t-1}+\ldots +\phi_{p}X_{t-p}+\epsilon_{t},$$where $\phi_{1},\phi_{1},\ldots ,\phi_{p}$ are autoregressive coefficients, *c* is a constant $\epsilon_{t}$ is the random error (shock or innovation) at time *t*.-The Integrated (I) component of the ARIMA model refers to the number of differencing operations that are needed to achieve data stationarity (the series’ statistical properties are constant over time). In the ARIMA model, *d* represents the order of differencing needed to achieve stationarity. If *d* is the order of differencing, the transformed series can be obtained as seen in Equation (4)
(4)$$Y_{t}={\left(1-B\right)}^{d}X_{t},$$where *B* is the lag operator (backshift) and *d* is the degree of differencing-The Moving-Average (MA) component assumes that the current value of the time series can be expressed as a sum of past random error terms (or residuals). In an MA(q) model, *q* represents the number of lagged error terms included in the model. The general formula for the MA component is seen in the following Equation (5):
(5)$$X_{t}=c+\epsilon_{t}+\theta_{1}\epsilon_{t-1}+\theta_{2}\epsilon_{t-2}+\ldots +\theta_{q}X_{t-q},$$where $\theta_{1},\theta_{2},\ldots ,\theta_{q}$ are the moving-average coefficients.

An ARIMA (p,d,q) model is defined by the following three parameters:-p: autoregressive (AR) component order-d: differentiating (I) component order-q: moving-average (MA) component order

An ARIMA model combines these three components in a general model, which can be written as seen in the following Equation (6):(6)$$X_{t}=c+\phi_{1}X_{t-1}+\phi_{1}X_{t-1}+\ldots +\phi_{p}X_{t-p}+\epsilon_{t}+\theta_{1}\epsilon_{t-1}+\theta_{2}\epsilon_{t-2}+\ldots +\theta_{q}X_{t-q},$$
where the $X_{t}$ series was differentiating *d* times in order to make it stationary

Among the various possible and validated models, the one with the smallest values of the Akaike Information Criterion (AIC) and the Schwarz Bayesian Information Criterion (BIC) is selected [59,60].

## 4. Results and Discussion

### 4.1. Average Length of Stay (ALoS)

The average length of stay (ALoS) has a general mean of 5.58 days over the analyzed period, with a maximum value of 6.08 recorded in the first quarter of 2011 and a minimum value of 4.97 calculated in the third quarter of 2023 (Figure 1). Thus, we can see that the data show a clear seasonal pattern, with higher values observed in the first and fourth quarters (winter-autumn) and lower values in the second and third quarters (spring-summer). This seasonality could be linked to factors such as the increased incidence of illnesses during the colder seasons (e.g., influenza, respiratory infections).

The standard deviation of the series is 0.24349, and the coefficient of variation is 4.36%, indicating a low level of variability around the mean, which implies that the series is homogeneous and that the mean is representative. The Shapiro–Wilk test returned a value of 0.99 with an associated *p*-value of 0.925, which means the data follow a normal distribution, indicating stability in the indicator’s behavior. The confidence interval for the mean is [5.516; 5.646].

As we can see in Figure 1, the average annual values of the ALoS remained relatively constant between 2010 and 2020, mostly around 5.6–5.7 days, with minor variations. However, after 2022, following the end of the COVID-19 pandemic, we can observe a decline in the ALoS, reaching its lowest value in 2023 (approximately 5.16 days), a fact that may reflect changes in post-COVID medical protocols or an increased efficiency in patient treatment and management.

To apply the ARIMA method for forecasting, we first checked whether the time series was stationary using the Dickey–Fuller test. According to Table 2, the *p*-value of the initial ALoS series is 0.9089, indicating that the series is not stationary, thus confirming the presence of a unit root. To address this, we apply the first differencing of the series, resulting in the differenced series D(ALoS).

According to the Dickey–Fuller test, the differenced series is stationary, as demonstrated by the associated *p*-value of zero (Table 3). This result suggests that the differenced series does not exhibit unit roots, making it suitable for ARIMA modeling.

Using the correlogram of D(AloS), we examined several potential ARIMA(p,1,q) models. Although multiple valid models were identified, the ARIMA(4,1,1) model was selected because of its lowest Akaike and Schwarz criterion values, indicating that it best explains the dynamics of the data. In this model, the estimated coefficients are statistically significant, as their *p*-values from the *t*-Statistic test are less than 0.05, and the F-test, with a value of 26.69 and a *p*-value of zero, further supports the overall validity of the model.

Additionally, the residuals of the model show no evidence of autocorrelation, as confirmed by the Durbin–Watson test statistic of 1.69 and the Breusch–Godfrey test, with an associated *p*-value of 0.30. These results suggest that the model captures the time-series dynamics accurately, with no significant autocorrelation among the residuals, which are independent of one another.

The ARCH test also supports the absence of heteroscedasticity, as the *p*-value of 0.34 is greater than the significance level of 0.05, as seen in Table 4. This indicates that the variance of the residuals remains constant over time, a crucial factor for the accuracy of the forecasts.

Furthermore, the ARIMA process is stationary and invertible, as demonstrated by the inverted roots of the AR and MA components being less than one, a necessary condition for making precise forecasts.

In Table 5, we can observe the forecast results, which are also visible in Figure 2, along with confidence intervals. Moreover, the historical data and the future forecast can be seen in Figure 1, highlighting the fit of the model and its predictive capability.

The ARIMA(4,1,1) model provides forecasts for the average length of stay (ALoS) over the next two years. As seen in Table 4, our projections show a continued decline in the ALoS, reaching a minimum of 4.95 days in the third quarter of 2024, followed by an increase during the autumn and winter quarters. In 2025, the trend is expected to stabilize, with values ranging between 4.92 and 5.07 days. The model suggests that, in the short term, the average length of stay will be around 5 days while still indicating a downward trend compared to previous years.

The ARIMA(4,1,1) can efficiently model the time series for average length of stay, capturing both seasonality and long-term trends while being able to adapt to data variations. The observed decline in the post-pandemic period likely reflects several systemic changes in the healthcare sector, and the model forecasts a slight continuation of this trend in the coming years.

### 4.2. Bed Occupancy Rate (BOR)

Analyzing Figure 3, we can observe the quarterly graphical representation of the bed occupancy rate, where a significant decline is showing starting from the second quarter of 2020, which might be explained by the restrictions and the reorganization of the healthcare system that happened during the COVID-19 pandemic. From the second half of 2022 onwards, the BOR returned to pre-pandemic levels, indicating a normalization of the hospital resources use.

Moreover, we can observe a seasonal pattern in the data, with higher occupancy rates in autumn and winter, periods associated with increased demand for medical care, and lower rates in spring and summer. The mean of BOR during the analyzed period is 62.3%, with a standard deviation of 14.68 and a coefficient of variation of 23.56%, which reflects moderate data variability, showing that while the values are somewhat dispersed around the mean, the mean remains representative.

The Shapiro–Wilk test returned a *p*-value of 0.005, meaning that the data do not follow a normal distribution. This suggests that the distribution of bed occupancy rates is influenced by certain factors, such as seasonal changes and the impact of unpredictable events such as the pandemic.

To determine whether the initial time series is stationary, we applied the Dickey–Fuller test using a model with both a constant and trend component. The test produced a *p*-value of 0.155, which is greater than the significance level of 0.05, as seen in Table 6. Thus, we accepted the null hypothesis, indicating that the initial series is not stationary and confirming the presence of a unit root.

To address this issue, we proceeded to differentiate the data, resulting in a new series, D(BOR), which was found to be stationary, as seen in Table 7. The *p*-value from the Dickey–Fuller test for the differenced series is 0.00, suggesting that the data are now suitable for ARIMA modeling.

Using the correlogram of D(BOR), we tested several possible ARIMA(p,1,q) models, out of which we selected the ARIMA(2,1,2) model due to its lowest Akaike and Schwarz criterion values. This choice indicates that the selected model is the most appropriate for capturing the dynamics of the bed occupancy rate (BOR) time series. As shown in Table 8, the estimated coefficients of this model are statistically significant, with the *p*-value being less than 0.05. This implies that the coefficients contribute meaningfully to modeling the bed utilization rate. The F-test for the model, with a value of 21.58 and a *p*-value of zero, further supports the overall validity of the model.

The Durbin–Watson test statistic of 2.12 and the Breusch–Godfrey test with a *p*-value of 0.64 confirm that the residuals are not autocorrelated, indicating that the model is well-fitted to the analyzed data and effectively captures the series’ dynamics without autocorrelation issues. Furthermore, the model is homoscedastic, as the ARCH test results in a *p*-value of 0.41, exceeding the 0.05 significance level. This suggests a constant error variation, which is essential for the model’s reliability in future forecasts. Additionally, the model is stationary and invertible, as the inverted roots of the AR and MA components are less than 1, ensuring the model’s stability for long-term forecasting.

Table 9 shows the forecast results, which are also presented, along with confidence intervals in Figure 4. The ARIMA(2,1,2) model forecasts the evolution of the BOR for 2024 and 2025, showing a gradual decline in this indicator during the warmer periods (spring and summer), followed by a slight increase in autumn and winter. The same seasonal patterns are expected in 2025, with BOR varying between 46.35% in the third quarter and 51.22% in the first quarter.

The time series for the BOR is well described by the ARIMA(2,1,2) model, which is able to capture both seasonality and long-term trends. The sharp decline in this indicator during the pandemic was temporary, the model forecasting stable values in the 46–52% range for following years, with typical seasonal variations. This analysis provides valuable insights for healthcare resource planning and indicates a return to stable occupancy patterns in the post-pandemic world.

### 4.3. Number of Cases (NC)

As we can see in Figure 5, the NC registers a moderate downward trend, with a sharp decline in the second quarter of 2020 due to the restrictions imposed by the COVID-19 pandemic. After 2022, as restrictions were relaxed and the pandemic was officially over, we can observe a gradual ascending trend, but without reaching the pre-pandemic highs. The indicator registers moderate seasonal variations, decreasing in the second and third quarters and increasing in the first and last quarters of every year, mirroring the trend of the BOR, where seasonal and climatic influences affect the medical services demand.

The average number of treated patients during the analyzed period is 13,804, with a standard deviation of 3064, indicating moderate fluctuations. The coefficient of variation is 22.19%, reflecting a relatively high variability around the mean, largely due to anomalies introduced by the pandemic, though the mean remains a meaningful metric. The Shapiro–Wilk test suggests that the data do not follow a normal distribution, with an associated *p*-value of zero, confirming that the time series exhibits deviations from normality, likely due to the shocks caused by the COVID-19 pandemic.

Before using ARIMA for forecasting, we use the Dickey–Fuller test in order to check the stationarity of the time series. As we can see in Table 10, the test returned a *p*-value of 0.513, which, since it is larger than the significance threshold of 0.05, validates the null hypothesis and, thus, confirms the presence of a unit root, which means the data are not stationary.

We then proceed to differentiate the data, obtaining a new time series, D(NC), for which the Dickey–Fuller test returned a *p*-value of zero, as seen in Table 11, which means the new data are stationary and thus fit to be used in the ARIMA model.

Using the correlogram of D(NC) we tested several potential ARIMA models, ARIMA(2,1,2) emerging as the most representative, according to the Akaike and Schwarz criteria. In the selected model, the estimated coefficients are statistically significant, with *p*-values from the t-Statistic test being less than 0.05 (Table 12), indicating that these coefficients significantly contribute to explaining the variability in the number of cases.

The residuals of the model are independent, as confirmed by the Durbin–Watson test statistic of 2.02 and the Breusch–Godfrey test with a *p*-value of 0.99. This suggests that the model effectively captures the time-series dynamics without undesirable autocorrelation effects. The ARCH test indicates the absence of heteroscedasticity, with a constant error variance, as the associated *p*-value is 0.33, well above the significance threshold of 0.05.

Moreover, the model is both stationary and invertible, ensuring stability and accuracy for long-term forecasts. The forecast results are presented in Table 13 and illustrated graphically with confidence intervals in Figure 6. The actual data and forecast are shown together in Figure 5.

The ARIMA(2,1,2) model allowed us to generate forecasts of NC for 2024 and 2025, and the results showed a stable trend with moderate seasonal variations, similar to the historic trends. Thus, the ARIMA(2,1,2) model is well-suited for capturing the dynamics of the number of cases, which show moderate seasonality and a slight increase as the effects of the pandemic diminish. Although the initial series was non-stationary, applying the first differencing stabilized the data, resulting in a valid model with relatively stable forecasts for 2024–2025.

### 4.4. Case Mix Index (CMI)

As we can see in Figure 7, between 2010 and 2023, the case mix index (CMI) registers several fluctuations but stabilizes on a moderate upward trend starting with 2019, while a sizeable decline appears in the second quarter of 2022. This decline is likely caused by the COVID-19 pandemic, which temporarily reduced case complexity since most complicated but non-urgent procedures were postponed.

The average value for the analyzed period is 1.24, with a standard deviation of 0.133 and a coefficient of variation of 10.71%, indicating moderate fluctuations in case complexity. This low variation means less variability around the mean compared to other time series, making this indicator more stable than NC. The Shapiro–Wilk test returned a *p*-value of zero, meaning that the data do not follow a normal distribution, probably due to variations generated by the COVID-19 pandemic.

Before using the ARIMA forecasting model, we tested its viability by checking the time series’ stationarity with the Dickey–Fuller test with a constant component. As we can see in Table 14, the *p*-value is greater than the threshold of 0.05 (0.2069), which leads to the validation of the null hypothesis, which states that the data are not stationary. This indicates the presence of a trend or long-term variation that must be removed before ARIMA models can be applied properly.

In order to prepare the data for the proper application of the ARIMA model, we proceeded to differentiate the time series, which resulted in a new series, D(CMI), which proved to be stationary, with a *p*-value of 0.0001, as it can be seen in Table 15. Thus, the null hypothesis is rejected, confirming the stationarity of the differenced series and validating the CMI data for future modeling and forecasting.

Analyzing the correlogram of the D(CMI) series, we explored several possible ARIMA(p,1,q) models, out of which we have chosen ARIMA(3,1,3), as it had the lowest values for both the Akaike and Schwarz criteria, indicating that this model is the best fit for the data.

As we can see in Table 16, the model’s estimated coefficients are statistically significant at the 5% significance level, with *p*-values from the t-Statistic test being less than 0.05. This suggests that the coefficients can explain the fluctuations in the CMI indicator.

The Durbin–Watson test returned a value of 2.31, and the *p*-value of the Breusch–Godfrey test registered a value of 0.55, which confirmed that the model’s residuals show no signs of autocorrelation. Moreover, the ARCH test’s *p*-value of 0.35 confirms the absence of heteroscedasticity, indicating a constant variance of the errors over time, with the ARIMA model being stationary and invertible, thus ensuring its stability and accuracy in long-term forecasts.

The CMI’s forecasts for 2023 and 2024 are presented in Table 17 and shown, along with confidence intervals, in Figure 7 and together with the historical data in Figure 8. Similarly with the other indicators, the ARIMA(3,1,3) model generated stable and accurate forecasts for CMI, both for short- and medium-term projections. Even if the initial series was non-stationary, the data were validated for being used in the model after first-order differencing.

The forecasts suggest that the CMI will be constant in 2024 and 2025, with minor fluctuations around a value of approximately 1.55. This stability reflects a normalization of healthcare practices following the pandemic’s disruptions, offering a reliable basis for capacity planning and resource allocation.

### 4.5. Average Cost per Hospitalization (ACH)

Figure 9 presents the evolution of the ACH between 2010 and 2023 and the forecasts for 2024 and 2025, showing a moderate upward trend until mid-2019, followed by significant fluctuations, especially in each year’s final quarter. These variations may be caused by the COVID-19 pandemic as well as additional end-of-year expenses.

The mean value of this indicator for 2010–2023 is 1010.28 RON, with a very high standard deviation of 1056.5 RON and a coefficient of variation of 104.5%, which implies that the mean is not representative due to the large variations caused by the COVID-19 pandemic and inflation. Additionally, the data do not follow a normal distribution, as indicated by the Shapiro–Wilk test, which yielded a *p*-value of zero, which suggests that the time series includes extreme values, probably influenced by the national economic situation as well as the pandemic, which have contributed to a spike in hospitalization costs.

In order to assess the series stationarity, we used the Dickey–Fuller test, which returned a *p*-value of 0.5814, as seen in Table 18, validating the null hypothesis, which means that the initial series is not stationary. We then proceeded to calculate the first-order difference, which resulted in a new series, D(ACH), which is stationary, as confirmed by the Dickey–Fuller test with a *p*-value of 0.0065 (Table 19). This allows us to proceed with ARIMA modeling.

Using the correlogram of D(ACH) and testing several possible ARIMA(p,1,q) models, ARIMA(4,1,1) was selected as the most suitable for forecasting. This choice was based on the lowest Akaike and Schwarz criteria values among several valid models. The significant AR(4) coefficient indicates long-term autocorrelation, while the MA(1) coefficient captures short-term variations through a moving-average component.

As seen in Table 20, the R-squared of 0.746 suggests that the model explains approximately 74.6% of the ACH’s variation, which indicates a good fit. Moreover, the Durbin–Watson test statistic of 1.81 suggests that there is no first-order autocorrelation of the residuals, while the Breusch–Godfrey test confirms the absence of higher-order autocorrelation, with a *p*-value of 0.30. In addition, the ARCH test indicates no heteroscedasticity, with a *p*-value of 0.5, while the ARIMA model is both stationary and invertible, making it suitable for accurate forecasting.

The forecasts suggest a significant increase in average costs, particularly in the fourth quarter of each year, as seen in Figure 10, likely influenced by the same factors observed in previous years (e.g., pandemic effects, inflation). This substantial increase in the fourth quarter follows a similar pattern to previous years and can be attributed to seasonal factors, such as additional expenditures in the healthcare sector at the end of the year.

In Table 21, we can see the forecast of the ACH for 2024 and 2025, which is also shown in Figure 10, along with confidence intervals, and in Figure 9, along with the historical data. The ARIMA(4,1,1) model provides a reliable forecast for the ACH in 2024 and 2025, anticipating a significant rise in hospitalization-associated expenses, especially during the final months of every year. Although the initial series was non-stationary, after stabilizing the data through differencing, we managed to develop a valid model that accounts for the various fluctuations and offers realistic predictions, accounting for the impact of the pandemic and inflation on medical costs. This analysis is valuable for financial planning and resource allocation in the healthcare system, enabling better preparation for expected cost increases.

We conclude this section with a brief presentation of the descriptive statistics of our variables, as shown in Table 22.

## 5. Conclusions

The average length of stay (ALoS) indicator showed a visible seasonality, with higher values during autumn and winter (Q1 and Q4) and lower values during spring and summer (Q2 and Q3), a pattern that can be correlated with an increased incidence of respiratory illnesses in the colder months. Moreover, the standard deviation of 0.24349 and the coefficient of variation of 4.36% suggest low variability in the data relative to the mean, which means the time series were stable and homogenous. In addition, the Shapiro–Wilk test confirmed that the data follow a normal distribution.

In the years that followed the end of the pandemic, our results showed a decline in the ALoS, which reached a low of 5.16 in 2023, a reduction that may reflect changes in medical protocols after COVID-19 or increased efficiency of the medical treatments. Out of several valid forecasting models, we chose ARIMA(4,1,1) as the most suitable for modeling the ALoS based on the minimum Akaike and Schwarz criteria values. This model has projected a continuous decline in the ALoS, which reached a minimum of 4.95 days in the third quarter of 2024, followed by a slight increase during the autumn and winter quarters. Regarding 2025, the ALOS is projected to fluctuate between 4.92 and 5.07 days, thus continuing the moderate downward trend compared to historical values.

The ARIMA(4,1,1) model was proven to be able to effectively capture the dynamics of the ALoS, both in terms of seasonality and long-term trends. This indicator’s steady decline after the pandemic may be proof of several improvements in the healthcare system, the model predicting a continuous downward trend in the next years. In conclusion, the ARIMA model indicates that, in the short term, the average length of stay will stabilize around 5 days, with a gradual declining trend reflecting changes within the post-pandemic healthcare system.

Starting from the second quarter of 2020, the bed occupancy rate (BOR) registered a clear descending trend, which can be linked to the restrictions and the reorganization of hospitals during the COVID-19 pandemic. However, after the second quarter of 2022, BOR gradually returned to normal levels, indicating stable occupancy rates similar to pre-pandemic times. Moreover, our findings showed specific seasonality for this indicator, with increased values in autumn and winter, when the demand for medical services is higher, and lower values in spring and summer, suggesting that the BOR is heavily influenced by seasonal factors, such as the rise in respiratory illnesses during the colder months.

Among the different possible ARIMA forecasting models, we have chosen ARIMA(2,1,2) since it was the most suitable for describing the data based on the minimum Akaike and Schwarz criteria values. Our forecasting model highlighted a continuation of the seasonal pattern, with a decrease of the BOR during the spring and summer, followed by an increase in autumn and winter. In 2024, the BOR is projected to range between 47.6% in the third quarter and 52.8% in the first, while for 2025, the BOR is expected to vary between 46.4% in the third quarter and 51.2% in the first, indicating stable values within the 46–52% range. This being said, our research has shown that the ARIMA(2,1,2) model can effectively capture the seasonality and medium-term trends of this indicator, reflecting a return to stable rates after the pandemic is over. Moreover, the model anticipates a stable rate of around 50%, with predictable seasonal variations, making it a valuable tool for healthcare resource planning.

Summing up, the ARIMA(2,1,2) model provided an accurate forecast of the bed occupancy rate for the following years, confirming the historic data’s post-pandemic stability, along with predictable seasonal fluctuations. This information can be instrumental for decisionmakers in optimizing hospital resource allocation and ensuring efficient healthcare service delivery.

Moving on to the third indicator, the number of cases (NC) exhibited a moderate decreasing trend from 2010 to 2023, with a more severe decline in the second quarter of 2020, when the pandemic started and the government restrictions were first put in place. After 2022, when the restrictions were lifted as the pandemic ended, the NC started to gradually increase, albeit with values lower than their pre-pandemic levels. In this case, as well, the data were moderately seasonal, with a lower number of cases during spring and summer and higher during autumn and winter, similar to the seasonal character of the bed occupancy rate. This seasonality can be explained by the rise in seasonal illnesses during the colder months.

Similar to the previous indicators, the Dickey–Fuller test confirmed that the data were non-stationary, hence the need for first-order differencing. After this operation, we managed to obtain a stationary dataset that was fit to be used in ARIMA forecasting. Among the different models, ARIMA(2,1,2) has proven to be the most suitable forecast for this indicator.

Our forecasting model predicted that the NC will stabilize in the following years, with moderate seasonal variations similar to those observed in the past. Moreover, the ARIMA model showed that the NC will range between 12,200 and 12,800 in 2024 and 2025, with slight increases in autumn and winter while maintaining stable post-pandemic trends. In addition, the ARIMA(2,1,2) model was able to effectively capture the seasonality and medium-term dynamics of this indicator, highlighting a steady recovery without fully returning to pre-pandemic levels. The application of first-order differencing allowed the data to be prepared for proper modeling and, thus, led to reliable forecasts for the next years.

Thus, our findings have shown that ARIMA(2,1,2) is a suitable model for forecasting the number of cases for both short- and medium-term periods, highlighting stability and moderate seasonal variations in the post-pandemic period. These insights can support the hospital in proper planning and allocating resources in order to be prepared for the project fluctuations.

Our fourth indicator, the case mix index, despite its moderate ascending trend since 2019, exhibited a significant decline starting with the second quarter of 2022. This was likely a consequence of the COVID-19 pandemic, during which the more complex but less urgent cases were postponed, temporarily reducing the CMI.

Among the different possible models, we selected ARIMA(3,1,3) as the most appropriate, based on the correlogram analysis and Akaike and Schwarz criteria indicating a good fit. According to our forecasting model, the CMI is projected to remain more or less constant in 2024 and 2025, with values in the 1.54–1.55 range, suggesting short- and medium-term stability without major fluctuations. Moreover, our findings showed that the ARIMA(3,1,3) model is robust and stable, offering reliable forecasts. Although the initial series was non-stationary, applying first-order differencing allowed us to obtain a suitable model that is able to capture the trends and moderate fluctuations of the post-pandemic CMI. Our forecast shows that this indicator is stable, which can be seen as a positive indicator for resource management and healthcare service planning.

In conclusion, out forecasting model was able to offer accurate and realistic projections for the case mix index, highlighting a consistent level of case complexity in the short to medium term. This stability is beneficial for healthcare administrators in optimizing resource allocation and consistent service delivery planning, indicating a return to normal and efficient medical management in the post-pandemic world.

Our last indicator, the average cost per hospitalization (ACH), showed moderate growth until mid-2019, followed by significant fluctuations, specifically sharp increases in the fourth quarter of each year. These seasonal increases appear to be related to more intense medical activities in the colder months, as well as increased end-of-year expenses, further amplified by the COVID-19 pandemic and inflation.

As with the other four KPIs, we used the correlogram analysis in order to find the most suitable ARIMA model, thus selecting ARIMA(4,1,1) based on the Akaike and Schwarz criteria. This model explained approximately 75.6% of the variation in average hospitalization costs, with the AR(4) and MA(1) coefficients indicating long-term autocorrelation and short-term variance adjustments.

Our forecast has shown that the ACH will increase significantly, especially in the final quarters of 2024 and 2025, reflecting the seasonal pattern observed in previous years. These projected increases, reaching 3203.20 RON in the fourth quarter of 2024 and 3029.85 RON in the fourth quarter of 2025, are attributed to seasonal factors and an economic context influenced by inflation. Thus, we can see that the ARIMA(4,1,1) model was able to realistically predict this indicator’s evolution, anticipating a continued upward trend, particularly in the final quarters of each year. These projections align with historical variations and reflect ongoing financial pressures in the healthcare system. This forecast can be useful for budget planning and resource allocation, highlighting the importance of managing the economic impact of seasonal and external influences on hospitalization costs.

Our findings have shown an increase in the CMI and the ACH, which aligns with trends identified in other scientific studies analyzing hospital resources and costs. For instance, Duarte et al. (2021) also predicted rising costs and increasing case complexity due to factors such as inflation and the strain on resources related to the pandemic. These findings are consistent with the forecast for Craiova, where the impact of inflation and the pandemic has significantly contributed to an upward trend in hospitalization costs.

Our findings showed a clear seasonality for the ALoS, the indicator increasing during the cold season (autumn and winter), a trend that can be correlated with a rise in respiratory illnesses. Observing this seasonality is important for effective medical resource planning, allowing for capacity optimization throughout the year. Our analysis also highlighted a reduction in the ALoS during the post-pandemic period, alongside a return of BOR to pre-pandemic levels. Based on this result, we can observe a need for systemic adjustments in treatment protocols, as well as optimizing resources and patient admissions in the post-COVID context, which is consistent with international observations.

In addition, our statistical model has managed to forecast significant increases in ACH, especially in the final quarters of the year, thus supporting the hospital’s prediction of its future financial burden. Moreover, our study identifies several influencing factors, such as inflation and increases in seasonal medical activities, correlated with the effects of the pandemic, which provide relevant information for financial decision-making and budget allocation.

The convergence of these trends suggests that the financial and operational pressures experienced at SCJU Craiova are part of broader systemic changes affecting healthcare institutions globally. The combined effects of increased case complexity, resource shortages, and economic factors have placed additional burdens on hospital budgets, necessitating adaptive strategies for effective resource management and cost control. This alignment with international studies reinforces the reliability of the ARIMA-based forecasts for Craiova, supporting the need for proactive financial planning in anticipation of continued cost increases driven by external pressures.

## Figures and Tables

**Figure 1 healthcare-13-00102-f001:**
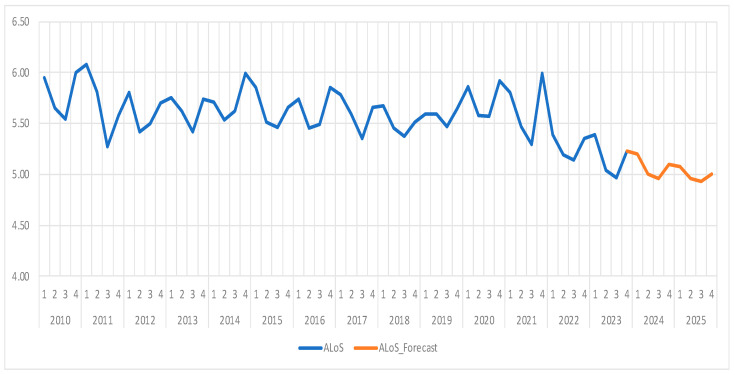
Average Length of Stay (ALoS) evolution and forecast. Own elaboration based on data from Appendix A.

**Figure 2 healthcare-13-00102-f002:**
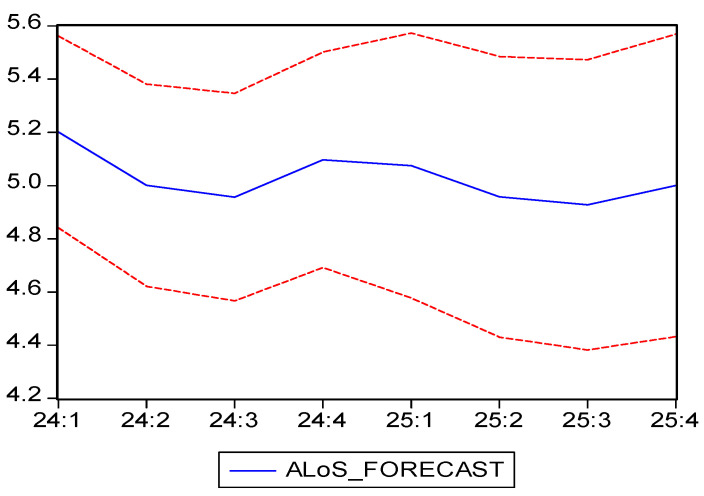
Average Length of Stay (ALoS) forecast and confidence interval (95%). Own elaboration based on data from Table 5.

**Figure 3 healthcare-13-00102-f003:**
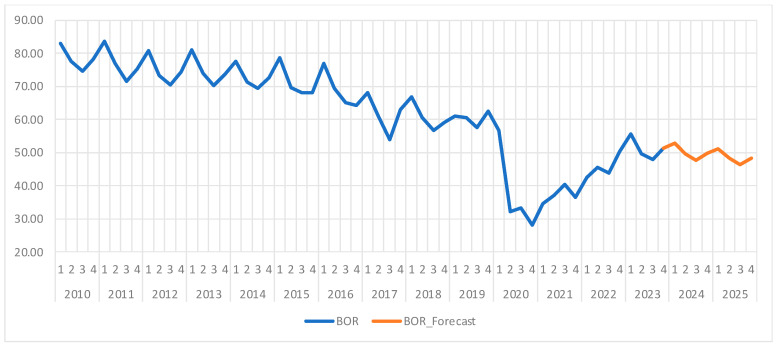
Bed occupancy rate (BOR) evolution and forecast. Own elaboration based on data from Appendix A.

**Figure 4 healthcare-13-00102-f004:**
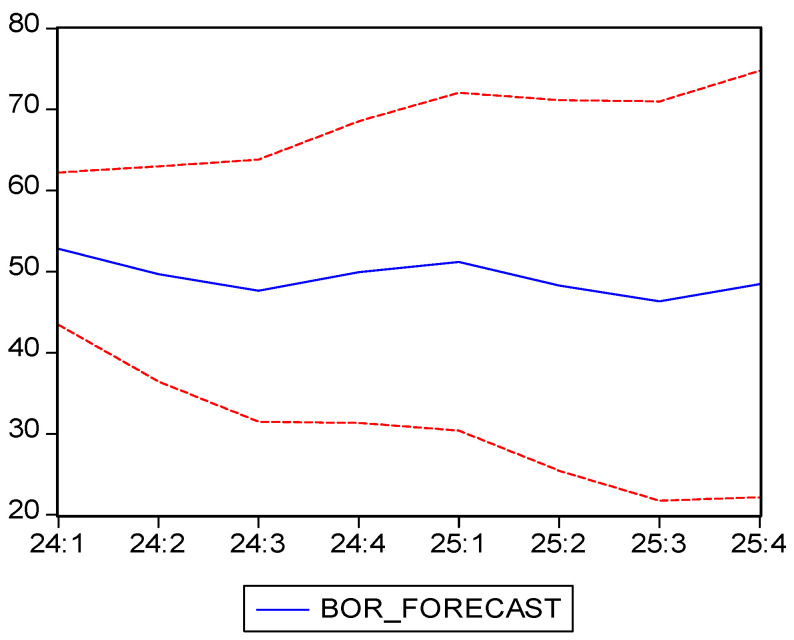
BOR forecast and confidence interval (95%). Own elaboration based on data from Table 8.

**Figure 5 healthcare-13-00102-f005:**
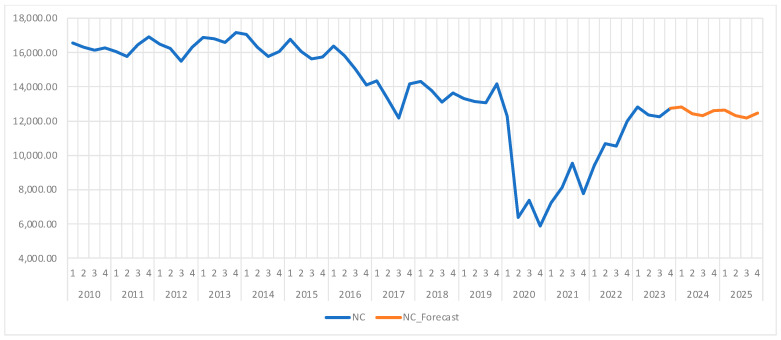
Number of cases (NC) evolution and forecast. Own elaboration based on data from Appendix A.

**Figure 6 healthcare-13-00102-f006:**
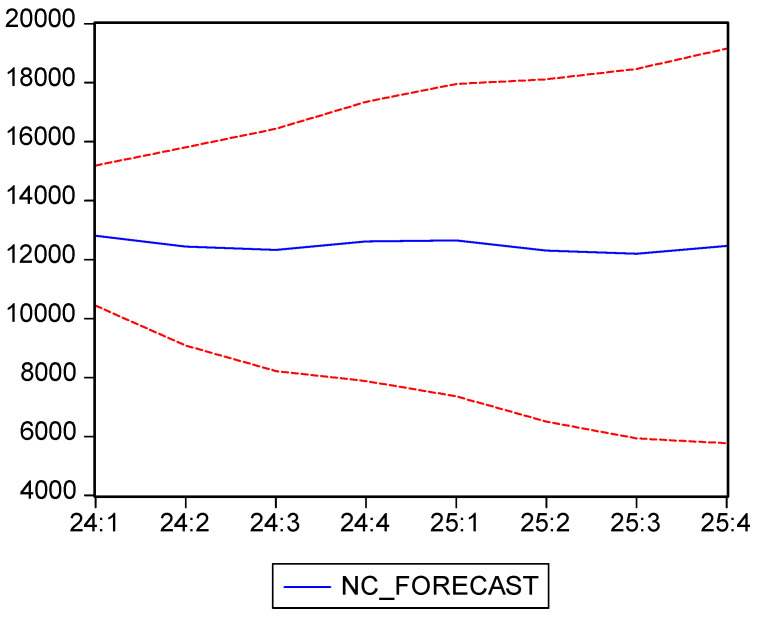
Number of cases (NC) forecast and confidence interval 95%. Own elaboration based on data from Table 12.

**Figure 7 healthcare-13-00102-f007:**
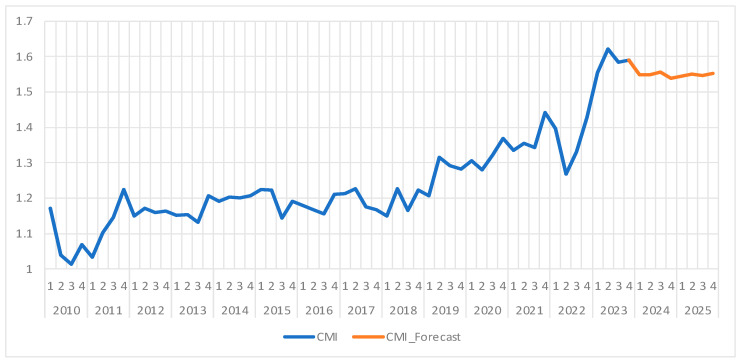
Case mix index (CMI) evolution and forecast. Own elaboration based on data from Appendix A.

**Figure 8 healthcare-13-00102-f008:**
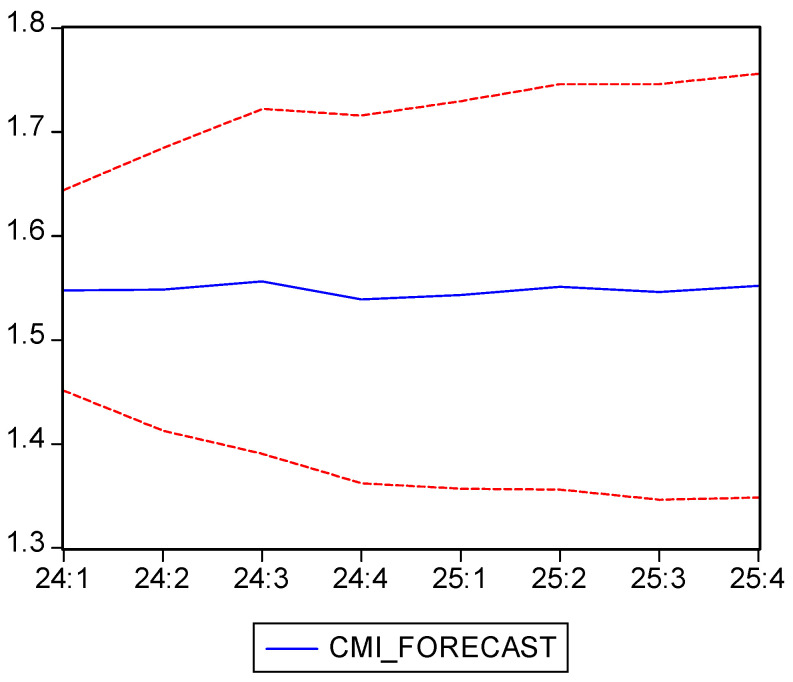
CMI forecast and confidence interval (95%). Own elaboration based on data from Table 16.

**Figure 9 healthcare-13-00102-f009:**
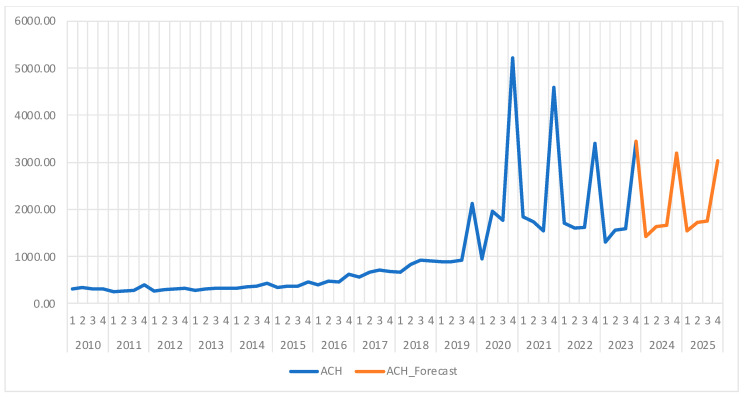
Average Cost per Hospitalization (ACH) evolution and forecast. Own elaboration based on data from Appendix A.

**Figure 10 healthcare-13-00102-f010:**
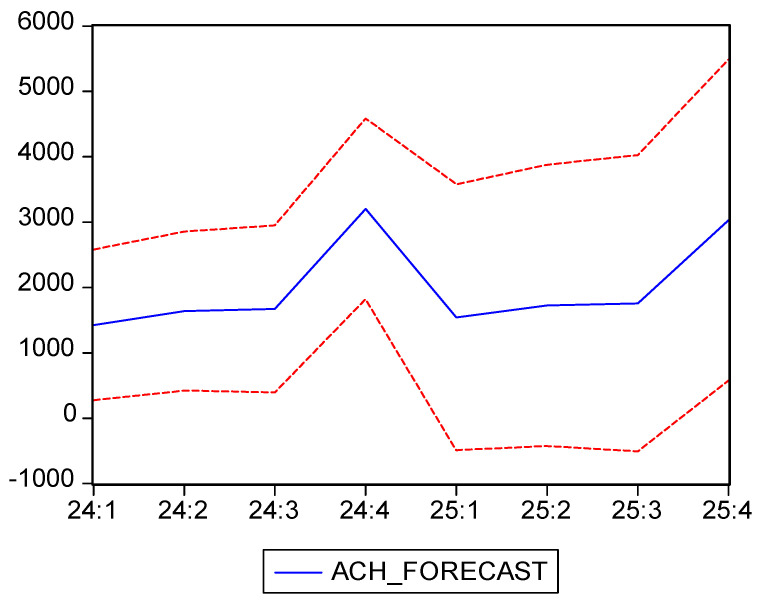
ACH Forecast and confidence interval (95%). Own elaboration based on data from Table 20.

**Table 1 healthcare-13-00102-t001:** Variable presentation.

No.	Variable Name	Variable Acronym	Type	Class	Data Continuity	Source
1	Average Length of Stay	ALoS	Discrete	Quantitative time series	2010–2023	Table A1
2	Bed Occupancy Rate	BOR	Discrete	Quantitative time series	2010–2023	Table A1
3	Number of Cases	NC	Discrete	Quantitative time series	2010–2023	Table A1
4	Case Mix Index	CMI	Discrete	Quantitative time series	2010–2023	Table A1
5	Average cost per Hospitalization	ACH	Discrete	Quantitative time series	2010–2023	Table A1

**Table 2 healthcare-13-00102-t002:** Dickey–Fuller test for ALoS.

Null Hypothesis: ALoS Has a Unit Root			*t*-Statistic	Prob.
Exogenous: Constant				
Augmented Dickey–Fuller test statistic			−0.354783	0.9089
Test critical values:	1% level		−3.562669	
	5% level		−2.918778	
	10% level		−2.597285	

**Table 3 healthcare-13-00102-t003:** Dickey–Fuller test for D(ALoS).

Null Hypothesis: D(ALoS) Has a Unit Root			*t*-Statistic	Prob.
Exogenous: Constant				
Augmented Dickey–Fuller test statistic			−0.354783	0.0000
Test critical values:	1% level		−3.562669	
	5% level		−2.918778	
	10% level		−2.597285	

**Table 4 healthcare-13-00102-t004:** ARIMA(4,1,1) model for D(ALoS).

Dependent Variable: D(ALoS)				
Variable	Coefficient	Std. Error	*t*-Statistic	Prob.
C	−0.012051	0.014908	−0.808377	0.4229
AR(4)	0.558266	0.117867	4.736386	0.0000
MA(1)	−0.757507	0.092254	−8.211122	0.0000
R-squared	0.526539	Mean dependent var		−0.016667
Adjusted R-squared	0.506812	S.D. dependent var		0.256185
S.E. of regression	0.179912	Akaike info criterion		−0.535677
Sum squared resid	1.553677	Schwarz criterion		−0.422040
Log-likelihood	16.65977	F-statistic		26.69059
Durbin–Watson stat	1.692857	Prob (F-statistic)		0.000000
Inverted AR Roots	0.86	0.00 + 0.86i	−0.00 − 0.86i	−0.86
Inverted MA Roots	0.76			
Breusch–Godfrey Serial Correlation LM Test:				
F-statistic	1.206241	Probability		0.308611
Obs × R-squared	2.428541	Probability		0.296927
ARCH Test:				
F-statistic	0.912608	Probability		0.344210
Obs × R-squared	0.932897	Probability		0.334111

**Table 5 healthcare-13-00102-t005:** ALoS forecast for 2024 and 2025.

Year/Quarter	Forecast (Days)
2024:1	5.20
2024:2	5.00
2024:3	4.95
2024:4	5.09
2025:1	5.07
2025:2	4.95
2025:3	4.92
2025:4	5.00

**Table 6 healthcare-13-00102-t006:** Dickey–Fuller test for BOR.

Null Hypothesis: BOR Has a Unit Root Exogenous: Constant, Linear Trend			*t*-Statistic	Prob.
Augmented Dickey–Fuller test statistic			−2.950668	0.1554
Test critical values:	1% level		−4.133838	
	5% level		−3.493692	
	10% level		−3.175693	

**Table 7 healthcare-13-00102-t007:** Dickey–Fuller test for D(BOR).

Null Hypothesis: D(BOR) Has a Unit Root Exogenous: Constant, Linear Trend			*t*-Statistic	Prob.
Augmented Dickey–Fuller test statistic			−7.721000	0.0000
Test critical values:	1% level		−4.140858	
	5% level		−3.496960	
	10% level		−3.177579	

**Table 8 healthcare-13-00102-t008:** ARIMA(2,1,2) model for D(BOR).

Dependent Variable: D(BOR)				
Variable	Coefficient	Std. Error	*t*-Statistic	Prob.
C	−0.363558	0.613830	−0.592277	0.5563
AR(2)	−0.962584	0.018233	−52.79228	0.0000
MA(2)	0.979296	0.029405	33.30356	0.0000
R-squared	0.463334	Mean dependent var		−0.435660
Adjusted R-squared	0.441868	S.D. dependent var		6.068875
S.E. of regression	4.533951	Akaike info criterion		5.916003
Sum squared resid	1027.836	Schwarz criterion		6.027529
Log-likelihood	−153.7741	F-statistic		21.58393
Durbin–Watson stat	2.125923	Prob (F-statistic)		0.000000
Breusch–Godfrey Serial Correlation LM Test:				
F-statistic	0.436324	Probability		0.648947
Obs × R-squared	0.839238	Probability		0.657297
ARCH Test:				
F-statistic	0.663497	Probability		0.419191
Obs × R-squared	0.681000	Probability		0.409242

**Table 9 healthcare-13-00102-t009:** BOR forecast for 2024 and 2025.

Year/Quarter	Forecast (%)
2024:1	52.84
2024:2	49.71
2024:3	47.64
2024:4	49.94
2025:1	51.22
2025:2	48.29
2025:3	46.35
2025:4	48.45

**Table 10 healthcare-13-00102-t010:** Dickey–Fuller test for NC.

Null Hypothesis: NC Has a Unit Root Exogenous: Constant			*t*-Statistic	Prob.
Augmented Dickey–Fuller test statistic			−1.529747	0.5113
Test critical values:	1% level		−3.555023	
	5% level		−2.915522	
	10% level		−2.595565	

**Table 11 healthcare-13-00102-t011:** Dickey–Fuller test for D(NC).

Null Hypothesis: NC Has a Unit Root Exogenous: Constant			*t*-Statistic	Prob.
Augmented Dickey–Fuller test statistic			−7.336749	0.0000
Test critical values:	1% level		−3.557472	
	5% level		−2.916566	
	10% level		−2.596116	

**Table 12 healthcare-13-00102-t012:** ARIMA(2,1,2) for D(NC).

Dependent Variable: D(NC)				
Variable	Coefficient	Std. Error	*t*-Statistic	Prob.
C	−35.42615	148.9963	−0.237765	0.8130
AR(2)	−0.960634	0.045277	−21.21687	0.0000
MA(2)	0.983871	0.030228	32.54782	0.0000
R-squared	0.091438	Mean dependent var		−63.66038
Adjusted R-squared	0.055096	S.D. dependent var		1185.370
S.E. of regression	1152.253	Akaike info criterion		16.99176
Sum squared resid	66,384,363	Schwarz criterion		17.10329
Log-likelihood	−447.2818	F-statistic		2.516018
Durbin–Watson stat	2.029202	Prob (F-statistic)		0.090962
Breusch–Godfrey Serial Correlation LM Test:				
F-statistic	0.009385	Probability		0.990661
Obs × R-squared	0.000000	Probability		1.000000
ARCH Test:				
F-statistic	0.955282	Probability		0.333084
Obs × R-squared	0.974868	Probability		0.323469

**Table 13 healthcare-13-00102-t013:** NC forecast for 2024 and 2025.

Year/Quarter	Forecast (Number)
2024:1	12,812
2024:2	12,440
2024:3	12,326
2024:4	12,613
2025:1	12,654
2025:2	12,308
2025:3	12,200
2025:4	12,462

**Table 14 healthcare-13-00102-t014:** Dickey–Fuller test for CMI.

Null Hypothesis: CMI Has a Unit Root Exogenous: Constant, Linear Trend			*t*-Statistic	Prob.
Augmented Dickey–Fuller test statistic			−2.790463	0.2069
Test critical values:	1% level		−4.133838	
	5% level		−3.493692	
	10% level		−3.175693	

**Table 15 healthcare-13-00102-t015:** Dickey–Fuller test for D(CMI).

Null Hypothesis: D(CMI) Has a Unit Root Exogenous: Constant, Linear Trend			*t*-Statistic	Prob.
Augmented Dickey–Fuller test statistic			−5.849725	0.0001
Test critical values:	1% level		−4.144584	
	5% level		−3.498692	
	10% level		−3.178578	

**Table 16 healthcare-13-00102-t016:** ARIMA(3,1,13) model for D(CMI).

Dependent Variable: D(CMI)				
Variable	Coefficient	Std. Error	*t*-Statistic	Prob.
C	0.007895	0.002981	2.648681	0.0108
AR(3)	0.511851	0.132208	3.871564	0.0003
MA(3)	−0.909709	0.044040	−20.65634	0.0000
R-squared	0.172550	Mean dependent var		0.009985
Adjusted R-squared	0.138777	S.D. dependent var		0.051179
S.E. of regression	0.047495	Akaike info criterion		−3.200416
Sum squared resid	0.110534	Schwarz criterion		−3.087844
Log-likelihood	86.21080	F-statistic		5.109046
Durbin–Watson stat	2.310610	Prob (F-statistic)		0.009653
Inverted AR Roots	0.80	−0.40 + 0.69i	−0.40 − 0.69i	
Inverted MA Roots	0.97	−0.48 + 0.84i	−0.48 − 0.84i	
Breusch–Godfrey Serial Correlation LM Test:				
F-statistic	0.594668	Probability		0.555843
Obs × R-squared	1.283346	Probability		0.526411
ARCH Test:				
F-statistic	0.862298	Probability		0.357648
Obs × R-squared	0.881973	Probability		0.347662

**Table 17 healthcare-13-00102-t017:** CMI forecast.

Year/Quarter	Forecast (Index)
2024:1	1.5476
2024:2	1.5484
2024:3	1.5562
2024:4	1.5389
2025:1	1.5432
2025:2	1.5510
2025:3	1.5460
2025:4	1.5521

**Table 18 healthcare-13-00102-t018:** Dickey–Fuller test for ACH.

Null Hypothesis: ACH Has a Unit Root Exogenous: Constant, Linear Trend			*t*-Statistic	Prob.
Augmented Dickey–Fuller test statistic			−2.011111	0.5814
Test critical values:	1% level		−4.148465	
	5% level		−3.500495	
	10% level		−3.179617	

**Table 19 healthcare-13-00102-t019:** Dickey–Fuller test for D(ACH).

Null Hypothesis: ACH Has a Unit Root Exogenous: Constant, Linear Trend			*t*-Statistic	Prob.
Augmented Dickey–Fuller Test Statistic			−4.308444	0.0065
Test critical values:	1% level		−4.148465	
	5% level		−3.500495	
	10% level		−3.179617	

**Table 20 healthcare-13-00102-t020:** ARIMA(4,1,1) model for D(ACH).

Dependent Variable D(ACH)				
Variable	Coefficient	Std. Error	*t*-Statistic	Prob.
C	35.56291	160.9220	0.220995	0.8260
AR(4)	0.826105	0.099550	8.298362	0.0000
MA(1)	−0.640525	0.111553	−5.741887	0.0000
R-squared	0.756462	Mean dependent var		62.49020
Adjusted R-squared	0.746314	S.D. dependent var		1058.558
S.E. of regression	533.1663	Akaike info criterion		15.45257
Sum squared resid	13,644,784	Schwarz criterion		15.56620
Log-likelihood	−391.0404	F-statistic		74.54715
Durbin–Watson stat	1.816762	Prob (F-statistic)		0.000000
Inverted AR Roots	0.95	0.00 − 0.95i	−0.00 + 0.95i	−0.95
Inverted MA Roots	0.64			
Breusch–Godfrey Serial Correlation LM Test:				
F-statistic	1.223978	Probability		0.303455
Obs × R-squared	2.576853	Probability		0.275704
ARCH Test:				
F-statistic	0.321865	Probability		0.573133
Obs × R-squared	0.333042	Probability		0.563873

**Table 21 healthcare-13-00102-t021:** The ACH Forecast for 2024 and 2025.

Year/Quarter	Forecast (RON)
2024:1	1426.70
2024:2	1638.58
2024:3	1671.20
2024:4	3203.20
2025:1	1543.72
2025:2	1724.94
2025:3	1758.07
2025:4	3029.85

**Table 22 healthcare-13-00102-t022:** Descriptive statistics.

No	KPIs	Mean	Standard Deviation	Variance	Kurtosis	Skewness	Maximum	Minimum
1	ALoS	5.58125	0.243490246	0.05928	−0.24349	−0.20293	6.08	4.97
2	BOR	62.30321	14.68268097	215.58112	−0.49277	−0.68053	83.74	28.06
3	NC	13,804.64	3064.160555	9,389,079.906	0.30331	−1.09007	17,148	5868
4	CMI	1.24205	0.133041809	0.0177001	1.40116	1.15012	1.6213	1.0142
5	ACH	1010.285	1056.532574	1,116,261.081	5.91512	2.32657	5226	256

## Data Availability

The original contributions presented in the study are included in the article. Further inquiries can be directed to the corresponding authors.

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
