# Peer review of "Statistical Analysis and Forecasts of Performance Indicators in the Romanian Healthcare System"

_healthcare, 2025, doi:10.3390/healthcare13020102_

Round 1
Reviewer 1 Report
Comments and Suggestions for Authors
The presented work stands out for its methodological solidity and the clarity with which it addresses the analysis and forecasting of hospital performance indicators. Among the main strengths, the rigorous and well-documented use of the models is highlighted, along with their careful validation through appropriate statistical tests. The consistency and reliability of the forecasts are well explained and assured. Another positive aspect is the clear and well-organized structure of the manuscript, which guides the reader through a logical progression from the introduction to the context, through the methodology, and finally to the results and their discussion. Furthermore, the literature used in the study is relevant and well-selected, providing solid theoretical support and adequately contextualizing the work within the framework of healthcare research.
However, there are some areas that could be improved to further enhance the quality of the manuscript:
• The results section, being detailed and rich in analysis, could be renamed “Results and Discussion.”
• The discussion could be expanded by adding a paragraph explaining why ARIMA models were chosen over alternative approaches, such as Machine Learning models, which could offer advantages in contexts with more complex dynamics. A reflection on this methodological choice would strengthen the scientific justification of the adopted model.
Author Response
Thank you for your review and for your valuable comments, which helped us enhance the quality of our manuscript. In the revised manuscript, we marked with red the additions that we have done based on your comments and the comments of the other reviewer. Below, you can find a detailed response to each of your comments.
- The results section, being detailed and rich in analysis, could be renamed “Results and Discussion.”
We have renamed the Results and Discussion section, line 390, marked with red in the revised manuscript
- The discussion could be expanded by adding a paragraph explaining why ARIMA models were chosen over alternative approaches, such as Machine Learning models, which could offer advantages in contexts with more complex dynamics. A reflection on this methodological choice would strengthen the scientific justification of the adopted model.
We have added the following paragraph in the introduction (lines 111-123, marked with red), explaining why ARIMA models were chosen over alternative approaches.
„The ARIMA model is based on well-founded statistical models, which provide a good interpretation of the data, and its parameters (autoregression, integration, moving average) have a clear meaning, which facilitates the understanding of the relationships in the time series. Due to its relatively easy-to-understand concepts, ARIMA can be implemented quickly and efficiently and is useful in interdisciplinary research, being more accessible to readers. ARIMA is very effective for univariate time series, with models based on autocorrelation and requires a relatively small data set for calibration, unlike machine learning models, which often require large volumes of data for robust results. The ARIMA model identifies moderate seasonalities in the analyzed data and the extended SARIMA model can be used in the case of high seasonality. Machine learning models can overcome the limitations of ARIMA in capturing complicated relationships or working with multivariate data, but this is not the case for the data analyzed in our work. Thus, the choice of ARIMA in this work is scientifically justified, offering high interpretability and efficiency.
Reviewer 2 Report
Comments and Suggestions for Authors
This study focuses on analyzing and forecasting key performance indicators (KPIs) for the District Emergency Clinic Hospital in Craiova, Romania. Average Length of Stay (ALoS), Bed Occupancy Rate (BOR), and other indicators were examined using ARIMA models with data from 2010 to 2023. The findings reveal that ALoS and BOR have seasonal effects, CMI indicates stability in case complexity, and ACH increases due to inflation. The study contributes to decision-making processes in healthcare management and provides essential insights into the effects of seasonality and economic factors. This study is important in terms of subject and method. For this study to be published, some deficiencies need to be addressed by the authors:
1. The study's abstract is well written, but some terms should be written as active rather than passive. For example, "We estimated ALOS to remain between 46-52%, reflecting seasonal changes."
2. The statements in the study's introduction should also be clear. Conditional or suggestion-containing statements should be included in the discussion or conclusion sections of the study. Example: Line 125-132
3. In the methodology section of the study, statements such as the type, class, data continuity, and source of the variables should be clearly stated. A table can be created for such data.
4. A table should be created for the descriptive statistics of the variable data. This table should include mean, standard deviation, variance, kurtosis, skewness, maximum, minimum, etc. This table should be included in the results section of the study. Otherwise, the explanations are insufficient.
5. What do the red lines in Figure 2 mean? Please provide legend information in the figures. The same should be applied to other figures.
6. The unit of the forecast data created for AloS should be written in Table 5. The same applies to other tables.
7. The study's result and conclusion sections were written too long. These sections should be shortened.
8. The conclusion section of the study should be reduced, and a discussion section should be created.
The work is well written, but the study sections should be well organized.
Author Response
Thank you for your review and for your valuable comments, which helped us enhance the quality of our manuscript. In the revised manuscript, we marked with red the additions that we have done based on your comments and the comments of the other reviewer. Below, you can find a detailed response to each of your comments.
- The study's abstract is well written, but some terms should be written as active rather than passive. For example, "We estimated ALOS to remain between 46-52%, reflecting seasonal changes."
We have reorganized the abstract, avoiding the passive voice (lines 18-20, 23, 24, 26, marked with red in the revised manuscript)
- The statements in the study's introduction should also be clear. Conditional or suggestion-containing statements should be included in the discussion or conclusion sections of the study. Example: Line 125-132
We have moved the parapgrahs from lines 125-132 to the conclusion section (lines 838-851) since they refer to our study’s findings.
- In the methodology section of the study, statements such as the type, class, data continuity, and source of the variables should be clearly stated. A table can be created for such data.
We have added a new table (Table 1 in the revised manuscript, lines 287-290, marked with red) which presents the five variables of our research, as well as their type, class, data continuity and source.
- A table should be created for the descriptive statistics of the variable data. This table should include mean, standard deviation, variance, kurtosis, skewness, maximum, minimum, etc. This table should be included in the results section of the study. Otherwise, the explanations are insufficient.
We have added Table 22 (lines 713-716), at the end of the results and discussion section, where we highlight the descriptive statistics of our variables, as per your recommendation.
- What do the red lines in Figure 2 mean? Please provide legend information in the figures. The same should be applied to other figures.
The red lines represent the confidence interval for the prediction, at a significance level of 5%. We have added this explanation in the figure description for Figures 2,4,6,8 and 10.
- The unit of the forecast data created for AloS should be written in Table 5. The same applies to other tables.
Thank you for this valuable input. We have added the measurement units for all variables in Tables 5, 9, 13, 17 and 21 of the revised manuscript, as well as Table A1, from the appendix.
- The study's result and conclusion sections were written too long. These sections should be shortened.
We reorganized the results section, renamed it results and discussion and reduced the conclusion sections, eliminating repetitive or redundant information.
- The conclusion section of the study should be reduced, and a discussion section should be created.
We have reduced the conclusion section by eliminating redundant information and we have renamed the results section „results and discussion”.
The work is well written, but the study sections should be well organized.
Thank you for your valuable inputs. We have tried to reorganize our manuscript according to your comments, while trying to preserve, at the same time, the logic and coehrence of the arguments/
Round 2
Reviewer 2 Report
Comments and Suggestions for Authors
Accept in present form
Author Response
Esteemed Mr.Bobby Jiang,
I am pleased to submit the revised version of our manuscript titled "Statistical Analysis and Forecasts on Performance Indicators in the Romanian Healthcare System" for consideration for publication in Healthcare.
We have included the phrase "confidence interval at a level of 95%" for Figure 2, 4, 6, 8 and 10.
At the same time, we added an acknowledgement at lines 873-875.